

# Accelerated construction of an *in vitro* model of human periodontal ligament tissue: vacuum plasma combined with fibronectin coating and a polydimethylsiloxane matrix

Wen Liao[1,6], Yoshiya Hashimoto[2], Yoshitomo Honda[3], Peiqi Li[4], Yang Yao[5], Zhihe Zhao[1] and Naoyuki Matsumoto[6]

[1] State Key Laboratory of Oral Diseases & National Clinical Research Center for Oral Diseases, Department of Orthodontics, West China Hospital of Stomatology, Sichuan University, Chengdu, Sichuan, China
[2] Department of Biomaterials, Osaka Dental University, Osaka, Japan
[3] Institute of Dental Research, Osaka Dental University, Osaka, Japan
[4] Department of Implantology, Osaka Dental University, Osaka, Japan
[5] State Key Laboratory of Oral Diseases & National Clinical Research Center for Oral Diseases, Department of Implantology, West China Hospital of Stomatology, Sichuan University, Chengdu, Sichuan, China
[6] Department of Orthodontics, Osaka Dental University, Osaka, Japan

Corresponding author
Yang Yao, yaoyang9999@126.com

## ABSTRACT

Tying shape memory wires to crowded teeth causes the wires to deform according to the dental arch. This deformation results in a resilient force that is delivered to the tooth. The appropriate amount of force can activate the osteogenetic and osteoclastic ability of the periodontal ligament (PDL) and the tooth can be moved. This is the biological basis of orthodontic treatment. To achieve further insight into the mechanisms underlying orthodontic treatment, we examined whether accelerated construction of an *in vitro* human PDL fibroblast (HPdLF) stretching model can be achieved by combining fibronectin coating and vacuum plasma treatment with polydimethylsiloxane (PDMS) cell-culture chambers. Each chamber was randomly assigned to a no-surface modification (NN), fibronectin coating (FN), vacuum plasma treatment (PN), or vacuum plasma treatment followed by a fibronectin coating (PF) treatment protocol. The physical and chemical features and ability to promote cellular proliferation of the PDMS chamber surfaces were evaluated. Cellular adhesion of four materials were evaluated and two best-proliferated groups were considered as better model-constructing surfaces and used in subsequent experiments and used in subsequent experiments. HPdLFs were cultured on these two kinds of chambers without stretching for 3 days, then with stretching for 7 days. Time-course gene expression cellular morphology were evaluated. Chambers in the PN group had high wettability and surface component changes. The FN and PF chambers had high cellular proliferation ability. They were selected into subsequent experiments. After 3 days of culturing HPdLFs on the PF and PN chambers, the cells in the PF chambers had significantly higher levels of runt-related transcription factor 2 (*Runx-2*) and osteocalcin (*OCN*) gene expression compared with the cells in the PN chambers. After cyclic stretch application to the cells in the PN and PF chambers, expression of the type-3 collagen (*COL-3*) gene in PF group continued to increase for 7 days and was significantly

higher than that in the PN group from day 5 onwards. The HPdLFs in the PF group showed parallel alignment from days 3 to 7 after imposition of cyclic stretch, while those in the PN group aligned in parallel from day 5 on. Our results suggested that applying a fibronectin coating to a PDMS chamber after plasma treatment can accelerate establishment of an *in vitro* PDL stretching model.

## INTRODUCTION

Human periodontal ligament (PDL) tissue is a well-organized and flexible soft tissue that attaches tooth roots to the alveolar bone (*Jonsson et al., 2011*). It is the only soft tissue present at the interface between the hard tissues of tooth and bone. It plays an important part in orthodontic treatment because it secretes cytokines upon deformation, which activates bone-specific transcription factors that induce activation of osteoclasts and osteoblasts and subsequent alteration of alveolar bone metabolism (*Jonsson et al., 2011*; *Liao et al., 2016*; *Liao et al., 2013*). In orthodontic treatment, dentists tie wires made with shape memory alloys to attachments bonded on the tooth surface. The resilient force of these wires is applied to the PDL and teeth are moved (*Antoun et al., 2017*). Orthodontic force is a complicated combination of stretching and compression forces, which results in directional orthodontic tooth movement (*Kanzaki et al., 2001*; *Kanzaki et al., 2004*; *Li et al., 2011*). The results of clinical observations suggest that bone formation is the primary consequence of stretching forces whereas bone resorption is the primary consequence of compression forces; however, in contrast to these clinical observations, researchers have found that the two forces have the same biochemical response (*Chen et al., 2015*; *Jonsson et al., 2011*). The similar biochemical responses to stretching and compression forces inspired us to further examine bone metabolism mechanisms in PDL tissues. To do so, it is important to establish a model of both stretching and compression forces to investigate the differences. An *in vitro* model of human PDL tissue that accurately mimics the PDL would shorten the gap between clinical observation and experimental findings. We successfully established an *in vitro* model of PDL compression previously, but it cannot be used in the study of PDL stretching (*Liao et al., 2016*; *Liao et al., 2013*).

To mimic the stretching characteristics of PDL, the ideal scaffold material should be non-toxic, cell-adhesive, flexible, and have a high tensile strength (*De Jong et al., 2017*). Polydimethylsiloxane (PDMS) is a non-toxic, chemically and biologically inert polymer with high tensile strength and structural flexibility and a low Young's modulus value (*Fitzgerald et al., 2019*). These features would seem to make PDMS an ideal scaffold material, but because it is hydrophobic, surface modification is required to permit adequate cellular adhesion (*Halldorsson et al., 2015*). Several chemical and physical treatment techniques have been used to modify PDMS surface properties (*Zhou et al., 2012*). Coating is a chemical treatment technique used to cover the material with hydrophilic molecules (e.g., fibronectin or Pluronic F127 surfactant). It enhances the binding of

extracellular matrix (ECM) components (e.g., collagen and fibrin) to the polymer (*Wu & Hjort, 2009*). Plasma treatment is a typical example of a physical treatment technique to enhance hydrophilicity. Different plasma preparations have varied efficacy for increasing hydrophilicity. *Zhou, Ellis & Voelcker (2010)* used Ar plasma followed by treatment with acrylic acid and found that this method enhances hydrophilicity, decontaminates the surface, and results in rearrangement of surface hydrophobic groups. A combination of chemical and physical surface modification was used by *Hattori, Sugiura & Kanamori (2010)*. The PDMS surface was first modified with polyacrylic acid using ultraviolet (UV) light, followed by coating with collagen and fibronectin. These were immobilized in a microarray format by UV-induced graft polymerization through a dehydration–condensation reaction, resulting in formation of identical spots of ECM proteins. CHO-K1 cells can be successfully cultured for two days on this surface, which is evidence of its hydrophilicity.

We combined vacuum plasma treatment and fibronectin coating to modify the PDMS cell-culture matrix surface for attachment of PDL cells. The results suggested that this method can be used to accelerate the establishment of an *in vitro* PDL tissue stretching model that mimics the bone metabolism associated with PDL stretch. Physical and chemical traits were assessed for each surface modification condition. Cellular proliferation, cytokine secretion, and cell polarity and gene expression under cyclic stretch conditions were tested.

## MATERIALS & METHODS

### PDMS chambers and surface modification protocols

PDMS in the shape of the chambers was purchased from Strex Inc., Osaka, Japan. Each chamber was square (side length, 16 mm; surface area, 256 mm$^2$). Each chamber was randomly assigned to one of four experimental groups that varied by surface treatment protocol. NN-group chambers did not undergo any surface modification and served as the control group. The other three chamber groups were the experimental groups. The chambers in group FN were treated with fibronectin by exposing the chamber to 0.02 mg/mL fibronectin (Wako, Osaka, Japan) for 30 min at 37 °C to attach the fibronectin to the surface. The surface was then washed three times with distilled water. A vacuum plasma treatment system (PC-35-OS, Strex, Osaka, Japan) was used to expose the chambers in group PN to a treatment voltage of 10 kV, current of 30 mA, and frequency of 10 kHz delivered in a vacuum for 10 s. The chambers in group PF were first treated with the vacuum plasma treatment system described above and were then exposed to fibronectin according to the protocol described for group FN.

### Physical and chemical features of the chambers

The PDMS chamber surface roughness was assessed using atomic force microscopy (AFM, SPM-9600; Shimadzu, Kyoto, Japan) at 25 °C. All images were acquired at 512 × 512 pixels, using a direct mode scan rate of 1 Hz. The topographic analyses were performed using silicone cantilever tips with a spring contact of 42 N/m and a resonance frequency of 300 kHz (OMCL-AC160TS-C2; Olympus, Tokyo, Japan). The scale range was 20 × 20 μm

$(x, y)$. The 10-point average roughness (Rz) value was determined as Rz $=$ (Yp1 $+$ Yp2 $+$ Yp3 $+$ Yp4 $+$ Yp5 $+$ Yv1 $+$ Yv2 $+$ Yv3 $+$ Yv4 $+$ Yv5)/5.

Before assessment of the chamber surfaces using a scanning electron microscope (SEM, S-4800, Hitachi, Tokyo, Japan), all chambers were pre-treated. The first step of the pre-treatment protocol was rinsing with Dulbecco's phosphate-buffered saline (PBS; Nacalai Tesque, Kyoto, Japan). The chambers were then fixed using 2% glutaraldehyde (Wako Pure Chemical Industries, Osaka, Japan) and post-fixed in 1% osmium tetroxide (Wako). Each chamber was then dehydrated using a series of increasing concentrations of ethanol in distilled water; *tert*-butyl alcohol was used for the final step (Kishida Chemical, Osaka, Japan). Following dehydration, the samples were sublimed in a *tert*-Butyl alcohol freeze dryer (VFD-21S; Vacuum Device, Mito, Japan) and sputter-coated with platinum using a coating device (Plasma Multi Coater PMC-5000; Meiwa, Tokyo, Japan). The SEM surface observation parameter was 5 kV and 5,000× magnification.

Chamber surface wettability was measured using a drop-shape analysis (DSA) system (DSA10Mk2; KRÜSS GmbH, Hamburg, Germany). To assess wettability, the surface of each PDMS test chamber was configured horizontally at a constant temperature of 25 °C. A 0.03-mL drop of distilled water was dropped vertically onto the chamber surface, and a photograph of the water drop on the chamber surface was obtained immediately. Three chamber samples were tested per treatment. For each sample, the water dropping was replicated ten times. Computer software included in the DSA system was then used to analyze the shape of each water drop and calculate the contact angle.

Surface atomic component changes were assessed using X-ray photoelectron spectroscopy (XPS) performed with a PHI X-tool system (PHI X-tool, Ulvac-Phi, Kanagawa, Japan) equipped with an Al–Kα radiation source (15 kV; 53 W; spot size: 205 μm). The pass energy was 112.00 eV, the step size was 0.100 eV, and the takeoff angle was 45°. Twenty scans were performed for each sample. The measurements were obtained for three randomly selected points on each sample, and the results were recorded as mean $\pm$ standard error (SE) values of the sample means for these three measurements.

## Cell seeding, proliferation, morphology, and genetic analysis

Human PDL fibroblasts (HPdLFs, CLCC-7049; Lonza, Basel, Switzerland) were cultured in a humidified incubator at 37 °C with 5% $CO_2$ in medium consisting of 85% Dulbecco's modified Eagle's medium, 10% fetal bovine serum, and 5% antibiotics (all from Nacalai Tesque). When the cells reached confluency, they were treated with 0.1% trypsin/EDTA (Nacalai Tesque) and 400 μL culture medium containing $6 \times 10^4$ cells fibroblasts from passages four to six were seeded onto each PDMS chamber under the same culture conditions.

HPdLF proliferation was measured for all four kinds of chambers using a cell counting kit (WST-8; Dojindo Molecular Technologies Inc., Kumamoto, Japan) according to the manufacturer's protocol. Briefly, at 1 h after seeding, the culture medium was extracted and each chamber was rinsed three times with PBS. Then, 400 μL cell counting medium (WST-8 reagent, 10-fold dilution in PBS) was gently added to each chamber. After culture in a humidified incubator at 37 °C for 120 min, 100 μL cell counting medium was transferred

| Table 1 | Pcr primer sequence. | | |
|---|---|---|---|
| Osteocalcin | Backward | 5′-GCCCAATACGACCAAATCC-3′ | |
| | Forward | 5′-CTTCCCTGTGCCTGTGTACC-3′ | |
| | Backward | 5′-TCAGCCGGTTGGTTTCTG-3′ | |
| Runx-2 | Forward | 5′-GGTTAATCTCCGCAGGTCAC-3′ | |
| | Backward | 5′-TGCTTGCAGCCTTAAATGACT-3′ | |
| COL3 | Forward | 3′-CTGGACCCCAGGGTCTTC-5′ | |
| | Backward | 5′-CATCTGATCCAGGGTTTCCA-3′ | |
| FGF2 | Forward | 5′-TTCTTCCTGCGCATCCAC-3′ | |
| | Backward | 5′-TGCTTGAAGTTGTAGCTTGATGT-3′ | |

into a 96-well plate and absorbance was measured at 450 nm using a micro-plate reader (Spectra Max M5; Molecular Devices, Sunnyvale, CA, USA). The optical density was defined as the proliferation value.

Cellular adhesion and cell morphology were examined using immunofluorescence microscopy at 1 h after the initial seeding. The chambers were rinsed three times with PBS, fixed with 4% formaldehyde in PBS for 20 min, again washed three times with PBS, shocked for 30 min in 0.2% Triton X-100 (ICN Biochemical, Aurora, OH, USA), blocked for 30 min in a blocking solution of bovine serum albumin (Sigma, St. Louis, Missouri, US), incubated in fluorescent staining solution (Alexa Fluor 488; Invitrogen Co., Carlsbad, CA, USA), placed in a 200-fold dilution of PBS, and incubated in a humidified incubator at 37 °C for 60 min. Following this preparation, the samples were washed three times with PBS and mounted on glass slides in a mounting medium containing 4-6-diamidino-2-phenylindole (DAPI Fluoromount-G; Southern Biotech, Birmingham, Alabama, USA). The cell sections were examined using a confocal laser scanning microscope (LSM-800; Carl Zeiss, Oberkochen, Germany).

Genetic analyses were performed for the cells seeded on the PF and PN chambers because these two types had the greatest cell proliferation. Total RNA was isolated from the cell-seeded PDMS chambers on days 1 and 3 after initial seeding (Qiagen, Hilden, Germany). Single-stranded cDNA was synthesized from the mRNA using High-Capacity RNA-to-cDNA Master Mix (Applied Biosystems, Foster City, CA, USA). Real-time polymerase chain reaction (RT-PCR) assay was performed using a Universal Probe Library Set, Human (Roche Diagnostics, Mannheim, Germany), a FastStart Universal Probe Master (Roche Diagnostics), and the two-stage program parameters on a Step One Plus PCR system (Applied Biosystems). The PCR conditions were 10 min at 95 °C, followed by 45 cycles of 15 s at 95 °C and 60 s at 60 °C. The primers used for PCR are presented in Table 1.

## Cellular polarity and genetic analysis under stretching conditions

Following the initial seeding, the cells were pre-incubated in the PDMS chambers for 3 days to achieve stable growth. Then, the culture medium was changed and the PDMS chambers underwent continuous cyclic stretch (6.25% for 10 cycles/min) for a total of 7 days using a stretching appliance (CS-1700; Strex, Osaka, Japan). Cellular polarity was observed using a phase-contrast microscope (IX70; Olympus, Tokyo, Japan) on days 0, 1, 3, 5, and 7

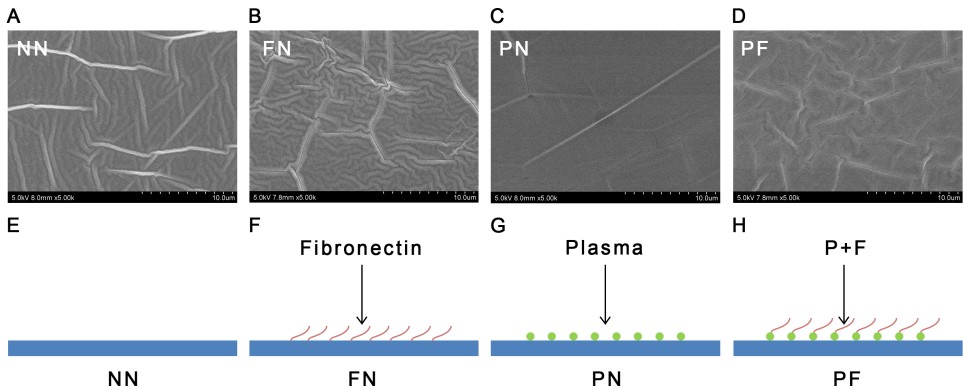

**Figure 1** **Scanning EM images (A–D) and diagrammatic sketches (E–H) and SEM images of four poly-dimethylsiloxane (PDMS) chambers with different surface modifications.** The group NN chambers showed most significant concave-convex profiles and groove-like structures on the surfaces. The group FN also showed significant concave–convex profiles. The group PF chambers had a concave–convex profile similar in size to that of group FN, but also somewhat similar to the PN profile, which was flatter in comparison. The group PN showed a flat surface. (SEM images, 5000X; NN, non-treated PDMS chamber; FN, fibronectin coated chamber; PN, plasma treated chamber; PF, chamber surface treated with plasma followed by coating with fibronectin; in E–H the blue flat surface indicates the chamber, red wavy line indicates long-chain fibronectin, green circle indicates the plasma treatment).

after the application of stretch. Images were obtained using a three-charge-coupled device (3CCD) digital camera (FX380; Olympus, Tokyo, Japan) and were digitally processed using image filing software (FlvFs; Flovel, Tokyo, Japan). Using the protocol described above, total RNA was isolated from the cell-seeded PDMS chambers at days 0, 1, 3, 5, and 7 after application of continuous cyclic stretch.

## Statistics

All results without test replicates were repeated three times. When possible, all results were presented as mean ± SD values. To compare the mean the four groups, one-way analysis of variance was conducted and if statistical significance was found, the method of LSD (least significance difference) was done as the post hoc analysis. For two groups comparison, non-pairing Student's t-tests was used. Test level was set to 0.05. SPSS 22.0 was used to conduct statistical analysis.

## RESULTS

### Physical characteristics of surface-modified PDMS chambers

The SEM results indicated that the group NN chambers had concave-convex profiles with many groove-like-structures on the surfaces that were approximately 1.0 μm wide. The group FN chambers also had concave-convex profiles and groove-like structures on the surfaces, but these grooves were smaller (i.e., narrower) than those in the group NN chambers. The group PN chambers had very flat surfaces. The group PF chambers had a concave-convex profile similar in size to that of group FN, but also somewhat similar to the PN profile, which was flatter in comparison (Fig. 1). The AFM results are presented in Fig. 2. Group PF had the greatest Rz value (40.204 nm), followed by group FN (Rz =

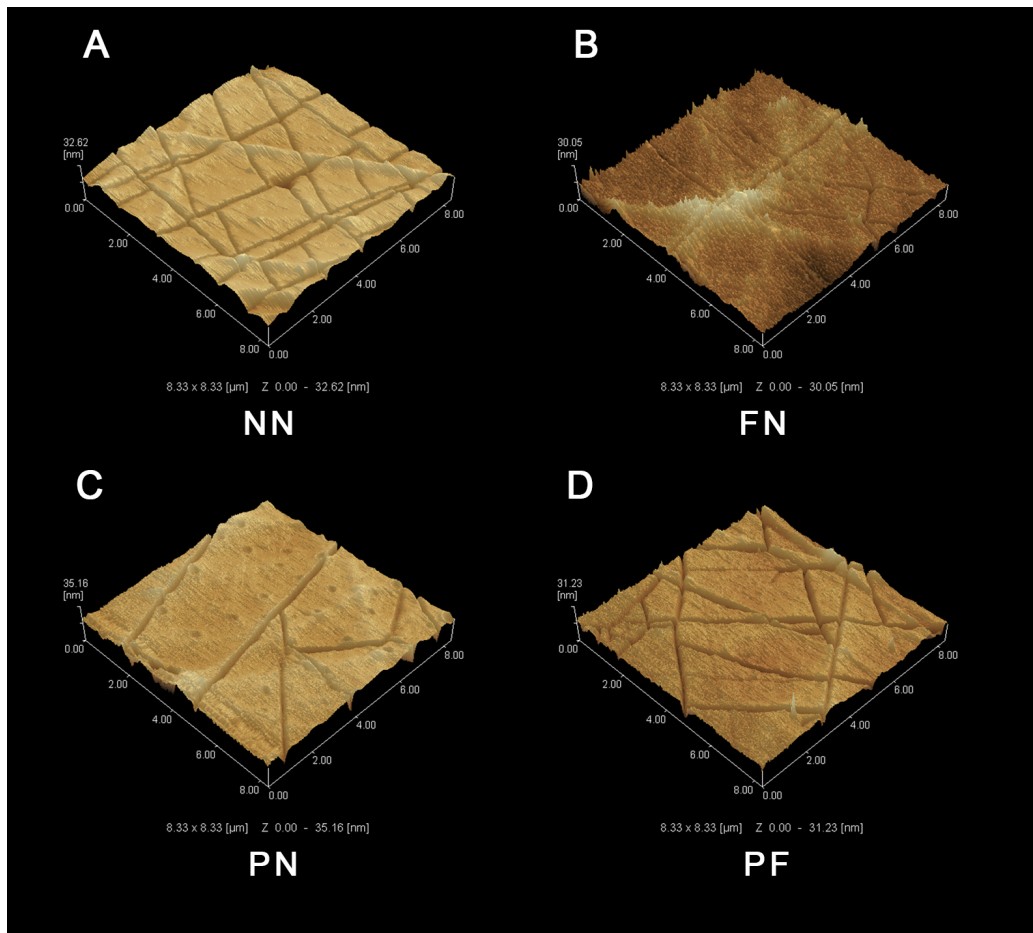

**Figure 2 Atomic force microscope (AFM) images of four polydimethylsiloxane (PDMS) chambers with different surface modifications.** (A) Group NN; (B) Group FN; (C) Group PN; (D) Group PF. Treatment group PF had the greatest Rz value, while control group NN had the lowest. Note that the difference between the PF and NN groups was quite similar to the sum of the difference between the FN and NN groups and the PN and NN groups.

37.272 nm), group PN (Rz = 34.878 nm), and the control group, NN (Rz = 32.629 nm). The results of the chamber surface wettability assessment are presented in Fig. 3. Group PN had the lowest distilled water-drop contact angle (64.6 ± 2.00°), compared with the chambers in the other groups, which had contact angles of 111.1 ± 0.19°, 107.6 ± 0.15°, and 109.3 ± 0.13°, for groups FN, PF, and NN, respectively ($P < 0.05$). Through the post hoc analysis, there was a statistical difference between each group (Table 2).

## Chemical characteristics of surface-modified PDMS chambers

The XPS results are presented in Fig. 4 and subsequent post hoc analysis results are shown in Table 3. The surface components of the group PN chambers were quite different than those of the other three groups. The carbon (C) proportion was 13.5 ± 2.5%, which was significantly lower than that of the other three groups ($P < 0.05$). The oxygen (O) proportion was 61.3 ± 2.2%, which was significantly higher than that of the other three

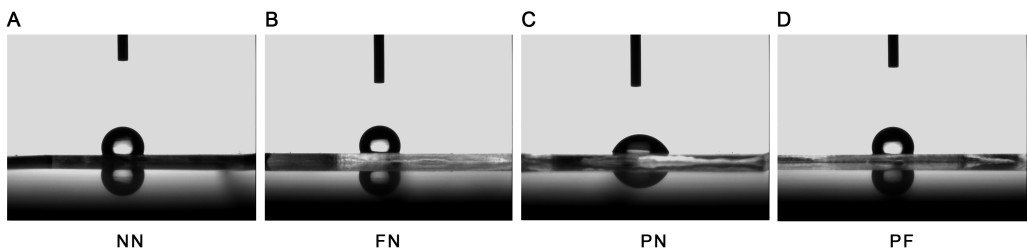

**Figure 3** **Wettability of four polydimethylsiloxane (PDMS) chambers with different surface modifications.** (A) Group NN; (B) Group FN; (C) Group PN; (D) Group PF. Only chamber PN had a contact angle that was significantly less than the control ($P < 0.05$). After plasma treatment and fibronectin coating (Chamber PF), the contact angle was very similar to chambers shown in chamber NN and FN, which were not treated with plasma.

**Table 2** Multiple comparisons of contact angles.

| (i) group | (j) group | Difference (i–j) | 95% Confidence interval | | $P$ |
|---|---|---|---|---|---|
| | | | Lower bound | Upper bound | |
| NN | FN | −1.782[*] | −1.918 | −1.645 | <0.001 |
| | NN | 44.764[*] | 44.627 | 44.900 | <0.001 |
| | PF | 1.745[*] | 1.609 | 1.882 | <0.001 |
| FN | NN | 1.782[*] | 1.645 | 1.918 | <0.001 |
| | NN | 46.545[*] | 46.409 | 46.682 | <0.001 |
| | PF | 3.527[*] | 3.391 | 3.664 | <0.001 |
| PN | NN | −44.764[*] | −44.900 | −44.627 | <0.001 |
| | FN | −46.545[*] | −46.682 | −46.409 | <0.001 |
| | PF | −43.018[*] | −43.155 | −42.882 | <0.001 |
| PF | NN | −1.745[*] | −1.882 | −1.609 | <0.001 |
| | FN | −3.527[*] | −3.664 | −3.391 | <0.001 |
| | NN | 43.018[*] | 42.882 | 43.155 | <0.001 |

**Notes.**
[*]$P < 0.05$.

groups ($P < 0.05$). Group FN had the highest C and lowest O proportion ($44.3 \pm 3.5\%$ and $33.0 \pm 3.6\%$, respectively); however, compared with the chambers in groups NN and PF, this difference was not significant. The chambers in group NN had C and O proportions of $33.4 \pm 4.3\%$ and $42.1 \pm 4.3\%$, respectively; group PF had C and O proportions of $39.9 \pm 6.0\%$ and $36.3 \pm 3.3\%$, respectively. The Si spectrum was fitted with two peaks associated with silicon bonding with carbon and oxygen (i.e., Si-($O_2$) at 102.1 eV or Si-C at 100.4 eV). Group PN had the highest percentage of Si-($O_2$); group PF had the highest percentage of C-Si (Fig. 4).

## Cellular proliferation on the surface-modified PDMS chambers

The results of the immunofluorescence assay for 1-hour cellular adhesion in the different chambers are shown in Fig. 5. Cell nuclei were stained blue and the cytoplasm (cytoskeleton) was stained green. Very few cells adhered to the chambers in group NN, whereas some

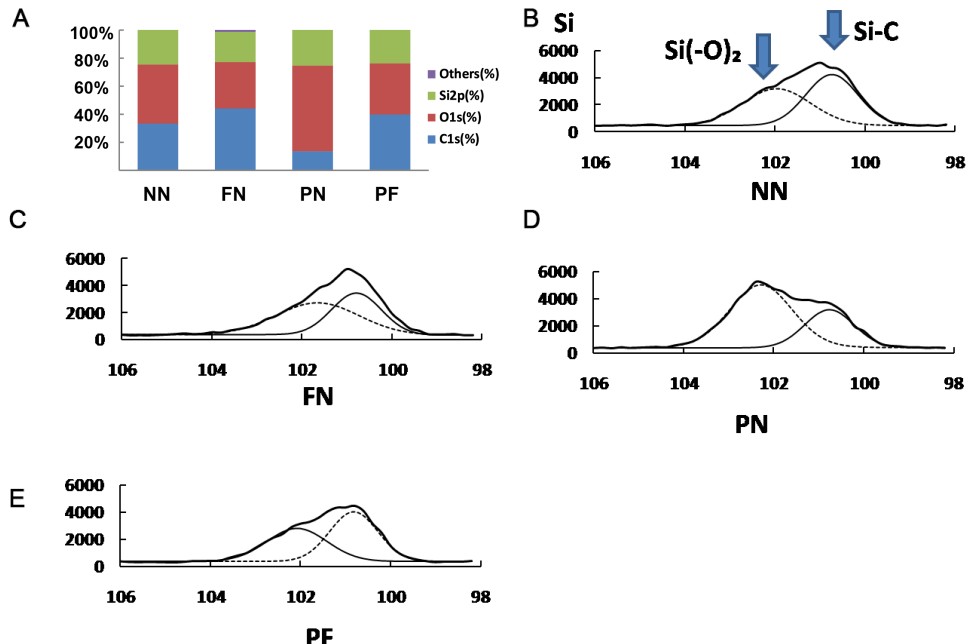

**Figure 4  X-ray photoelectron spectroscopy (XPS) showing surface elements components (A) and chemical bond analysis of Si for four polydimethylsiloxane (PDMS) chambers with different surface modifications (B–E).** The carbon (C) proportion of group PN was significantly lower than that of the other three groups ($P < 0.05$), while its oxygen (O) proportion was significantly higher than those ($P < 0.05$). Group FN had the highest C and lowest O proportion, which had no significance in difference with groups NN and PF.

staining of nuclei was found on the surfaces of chambers in group FN. The cytoplasm of these cells was contracted and gathered together; most cells had a very small area of cytoplasm, some cells were clumped together, rather than separate. This result indicated unbalanced, poor cellular adhesion. Compared with the cells from the FN chambers, the cells from the group PN chambers showed more green staining (i.e., more cytoplasm). This result indicated that the cellular adhesion was much better in the PN chambers than in the FN chambers. The cells from the group PF chambers had noticeably more cellular adhesion than the group FN cells, as well as more cytoplasm than the cells from the group PN chambers. These observations were quantified using WST-8 cell counting. The WST-8 results indicated that the group NN cells had the poorest cellular adhesion and the group FN cell adhesion was only slightly better than the NN cells. The group PN and PF cells had the best adhesion, and the differences between them were not statistically significant (Fig. 5). The post hoc analysis results was presented in Table 4.

Based on the cell proliferation results, the PN and PF groups were selected as the best model construction surfaces and were used for subsequent examination.

**Table 3  Multiple comparisons of carbon, oxygen, and Si proportion.**

| Dependent variable | (i) group | (j) group | Difference (i–j) | 95% Confidence interval | | $P$ |
|---|---|---|---|---|---|---|
| | | | | Lower bound | Upper bound | |
| C1s | NN | FN | −10.833* | −16.779 | −4.888 | .003 |
| | | NN | 19.967* | 14.021 | 25.912 | <0.001 |
| | | PF | −1.433 | −7.379 | 4.512 | .593 |
| | FN | NN | 10.833* | 4.888 | 16.779 | .003 |
| | | NN | 30.800* | 24.855 | 36.745 | <0.001 |
| | | PF | 9.400* | 3.455 | 15.345 | .007 |
| | PN | NN | −19.967* | −25.912 | −14.021 | <0.001 |
| | | FN | −30.800* | −36.745 | −24.855 | <0.001 |
| | | PF | −21.400* | −27.345 | −15.455 | <0.001 |
| | PF | NN | 1.433 | −4.512 | 7.379 | .593 |
| | | FN | −9.400* | −15.345 | −3.455 | .007 |
| | | NN | 21.400* | 15.455 | 27.345 | <0.001 |
| O1s | NN | FN | 9.100* | 3.196 | 15.004 | .007 |
| | | NN | −19.200* | −25.104 | −13.296 | <0.001 |
| | | PF | 2.100 | −3.804 | 8.004 | .436 |
| | FN | NN | −9.100* | −15.004 | −3.196 | .007 |
| | | NN | −28.300* | −34.204 | −22.396 | <0.001 |
| | | PF | −7.000* | −12.904 | −1.096 | .026 |
| | PN | NN | 19.200* | 13.296 | 25.104 | <0.001 |
| | | FN | 28.300* | 22.396 | 34.204 | <0.001 |
| | | PF | 21.300* | 15.396 | 27.204 | <0.001 |
| | PF | NN | −2.100 | −8.004 | 3.804 | .436 |
| | | FN | 7.000* | 1.096 | 12.904 | .026 |
| | | NN | −21.300* | −27.204 | −15.396 | <0.001 |
| Si2p | NN | FN | 3.033* | 1.075 | 4.992 | .007 |
| | | NN | −.833 | −2.792 | 1.125 | .355 |
| | | PF | −.733 | −2.692 | 1.225 | .413 |
| | FN | NN | −3.033* | −4.992 | −1.075 | .007 |
| | | NN | −3.867* | −5.825 | −1.908 | .002 |
| | | PF | −3.767* | −5.725 | −1.808 | .002 |
| | PN | NN | .833 | −1.125 | 2.792 | .355 |
| | | FN | 3.867* | 1.908 | 5.825 | .002 |
| | | PF | .100 | −1.858 | 2.058 | .909 |
| | PF | NN | .733 | −1.225 | 2.692 | .413 |
| | | FN | 3.767* | 1.808 | 5.725 | .002 |
| | | NN | −.100 | −2.058 | 1.858 | .909 |

Notes.
*$P < 0.05$

## Gene expression of HPdLFs seeded in PN and PF chambers

Two early and middle bone metabolism-related genes (*Runx-2* and *OCN*) were assayed using RT-PCR 1 day and 3 days after culturing without stretching. Glyceraldehyde-3-phosphate dehydrogenase (*GAPDH*) was used as a control. On day 1 after cell seeding,

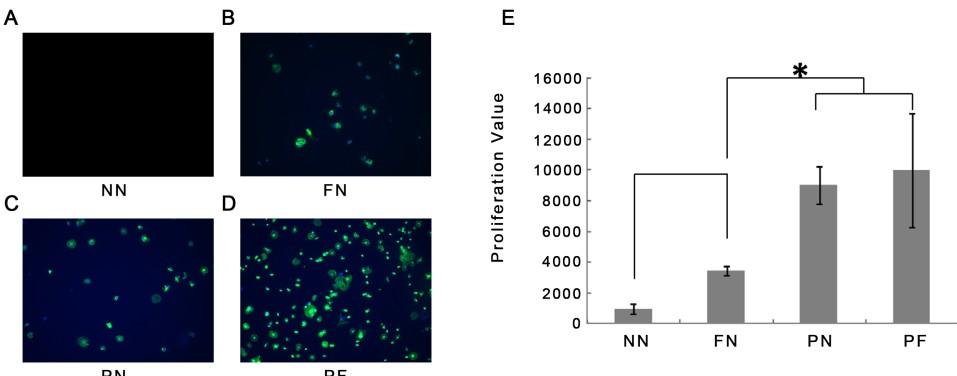

**Figure 5** Immunofluorescence microscopy observations 1 hour after cellular adhesion and proliferation of cells on four polydimethylsiloxane (PDMS) chambers with different surface modifications. (A) Group NN; (B) Group FN; (C) Group PN; (D) Group PF; (E) Proliferation value of four groups. Note that the fewest cells were associated with chambers NN. Chamber FN had more cells, but the number was still low. Chambers PN and PF had many more cells than chamber FN with statistic significance ($P < 0.05$). (E) shows that there is no significant difference between the cell numbers between groups PN and PF (blue, cell nuclei stain blue; green, the cytoskeleton stains).

**Table 4  Multiple comparisons of cellular adhesion.**

| (i) group | (j) group | Difference (i–j) | 95% Confidence interval | | $P$ |
| --- | --- | --- | --- | --- | --- |
| | | | Lower bound | Upper bound | |
| NN | FN | −2483.0 | −6166.3 | 1200.3 | .159 |
| | NN | −8050.5* | −11733.8 | −4367.2 | .001 |
| | PF | −9053.0* | −12736.3 | −5369.7 | <0.001 |
| FN | NN | 2483.0 | −1200.3 | 6166.3 | .159 |
| | NN | −5567.5* | −9250.8 | −1884.2 | .008 |
| | PF | −6570.0* | −10253.3 | −2886.7 | .003 |
| PN | NN | 8050.5* | 4367.2 | 11733.8 | .001 |
| | FN | 5567.5* | 1884.2 | 9250.8 | .008 |
| | PF | −1002.5 | −4685.8 | 2680.8 | .548 |
| PF | NN | 9053.0* | 5369.7 | 12736.3 | <0.001 |
| | FN | 6570.0* | 2886.7 | 10253.3 | .003 |
| | NN | 1002.5 | −2680.8 | 4685.8 | .548 |

**Notes.**
  *$P < 0.05$

both groups showed similar results. On day 3 after cell seeding, the cells from the group PF chambers had significantly higher levels of both *Runx-2* and *OCN* expression (Fig. 6) compared with the cells from the group PN chambers ($P < 0.05$).

## Cell polarity and gene expression after stretch application

Changes in orientation in the cells in the PN and PF groups started at day 3 after the start of stretching, and the group PF cells were already aligned almost parallel to each other on day 3. Similar parallel alignment occurred in the group PN cells on day 5. Both groups maintained parallelism until day 7. Application of stretching force had induced the

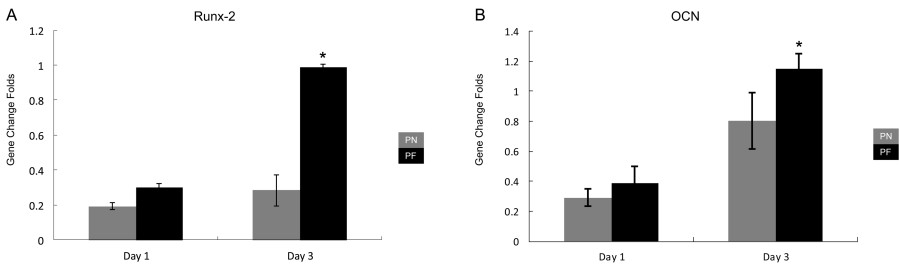

**Figure 6  Expression of early-bone metabolism-related gene expression after culturing in the poly-dimethylsiloxane (PDMS) chambers for 1 day and 3 days.** (A) *Runx-2* gene expression; (B) *OCN* gene expression. Note that cells from group PF had significantly higher *Runx-2* gene expression than group PN on day 1 and day 3, and *OCN* gene expression on day 1. The growth speed of group PF gene expression level is faster than PN group. ($P < 0.05$, *Runx-2*, runt-related transcription factor 2; *OCN*, osteocalcin; glyceraldehyde-3-phosphate dehydrogenase (*GADPH*) used as control).

alignment of most cells in a confluent state at an almost fixed angle that was relative to the stretch direction (Fig. 7).

The results for gene expression changes at 0, 1, 2, 3, 5, and 7 days after application of stretching are presented in Fig. 8. Three middle and late bone metabolism-related genes [*OCN*, *COL-3*, and fibroblast grow factor-2 (*FGF-2*)] were assayed using RT-PCR and with *GAPDH* as a control. The group PF cells had significantly higher expression levels of *COL-3* than the group PN cells at days 5 and 7. On day 0, *OCN* expression in the group PF cells was significantly higher than that of the PN group cells. The expression of *FGF-2* tended to increase from day 0 to day 3 and then slowly decreased; however, these changes with time were not significant (Fig. 8).

## DISCUSSION

### Physical and chemical basis for the good cellular proliferation abilities of groups PN and PF

PDMS is a hydrophobic material, and therefore chemical and physical treatments were devised to improve its surface wettability, which is usually considered conducive to cell and ECM growth (*Halldorsson et al., 2015*; *Ren et al., 2008*). The wettability results (Fig. 3) were consistent with the XPS results (Fig. 4), and group PN chambers had the greatest changes in both wettability and XPS characteristics. The chambers in this group had the highest wettability, highest O proportion, and lowest C proportion among all the chambers, and all the differences were statistically significant. The O proportion result indicated that the Si(-O$_2$) peak was much higher for the PN chambers compared with the other types; a hydrophilic silica-like layer had formed on the PDMS surface, which resulted in high wettability (*Razavi & Thakor, 2018*). The result that the lowest C proportion was in the PN chambers indicated that the methyl groups initially on the external surface of the chambers had moved to the internal surfaces (Fig. 4). This phenomenon can explain why the chamber surfaces changed from hydrophobic to hydrophilic. This result was consistent with *Larson et al. (2013)* and *Ko et al. (2015)* results. Surface area size is an important component of effective adhesion, because a larger surface area will have more area for

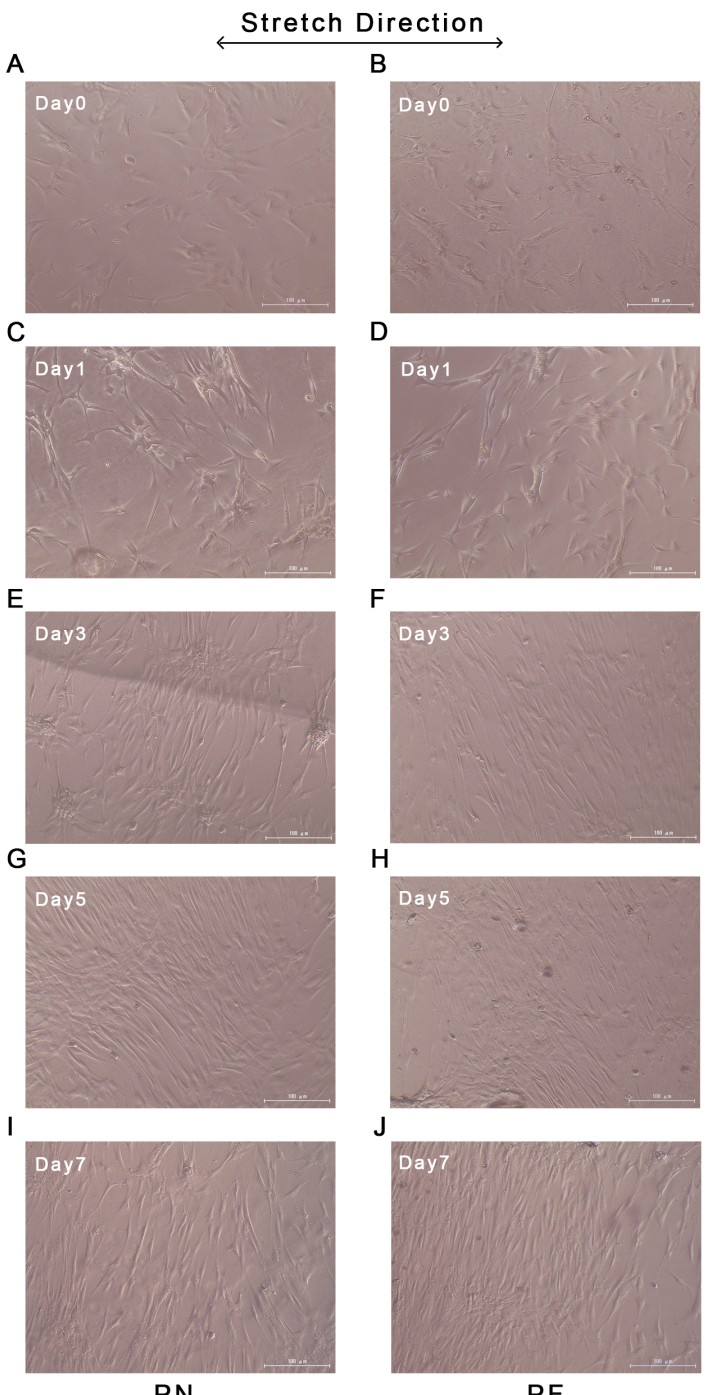

**Figure 7** **Cellular alignments during stretching of polydimethylsiloxane (PDMS) chambers with one of two surface modifications: either plasma treated (PN), or plasma treated followed by fibronectin (PF).** Group PF cells aligned almost parallel to each other on day 3, while similar parallel alignment occurred in the group PN cells on day 5. Both groups maintained parallelism until day 7. (A) Cellular alignment after 0 days of stretch in group PN; (B) cellular alignment after 0 days of stretch in group PF; (C) cellular alignment after 1 days of stretch in group PN; (D) Cellular alignment after 1 days of stretch in group PF; (E) cellular alignment after 3 days of stretch in group PN; (F) cellular alignment after 3 days of stretch in group PF; (G) cellular alignment after 5 days of stretch in group PN; (H) cellular alignment after 5 days of stretch in group PF; (I) cellular alignment after 7 days of stretch in group PN; (J) cellular alignment after 7 days of stretch in group PF.

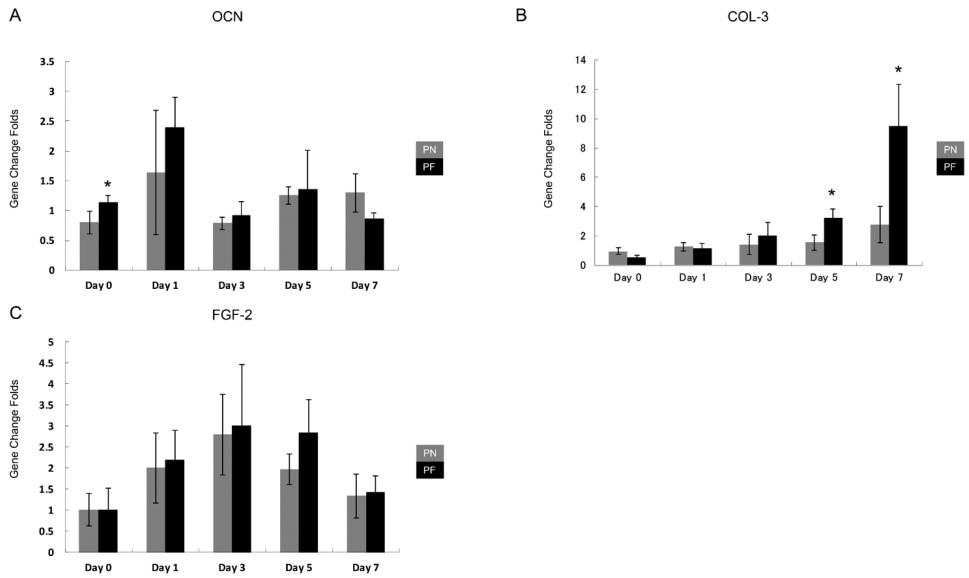

**Figure 8   Time course of expression of late-bone metabolism-related genes in cells during stretching in polydimethylsiloxane (PDMS) chambers with one of two surface modifications: either plasma treated (PN), or plasma treated followed by fibronectin (PF).** Note the *COL3* expression at days 5–7 in cells from group PF, which is consistent with the building of a much more mature periodontal ligament tissue than appears to be occurring in cells from group PN. (*OCN*, osteocalcin; *COL3*, collagen 3; *FGF2*, fibroblast grow factors-2; glyceraldehyde-3-phosphate dehydrogenase (*GADPH*) was used as control). (A) *OCN* gene expression; (B) *COL-3* gene expression; (C) *FGF-2* gene expression.

cellular adhesion (*Kermani et al., 2019*). All the chambers were of the same length and width, so the surface with the higher roughness had the greater surface area. The AFM results (Fig. 2) indicated that the 10-point average roughness of the group PF chambers was the highest among the four groups. Hence, the PF group chambers had the best cellular adhesion and subsequently, the best cellular proliferation. The difference between the FN and NN groups was approximately 4.5 nm, and the difference between the PN and NN groups was approximately 2 nm. The difference between the PF and NN groups was approximately 7.5 nm, which is similar to the sum of the first two differences. This result indicated that the plasma treatment and fibronectin coating did not interfere with each other and can be used together to improve the cell-culture performance of PDMS.

## Group PF's accelerated model construction ability, with or without stretch force application

The PF group's model construction ability was associated with the result that the PF and PN groups showed the best cellular adhesion. The PCR results for the HPdLFs seeded in the PN and PF chambers indicated that the expression of both the *Runx-2* and *OCN* genes was significantly higher in the PF group on day 3 ($P < 0.05$), but both groups had similar *Runx-2* and *OCN* gene expression on day 1. *Runx-2* is a key gene marker of HPdLF metabolism within bone, and the expression of this gene often serves as an early marker for osteogenic activity. *OCN* is an osteogenic marker gene that serves as a mid-range marker

of bone metabolism (*Oortgiesen et al., 2012*; *Yu et al., 2013*). Taken together, these results suggest that the PF treatment reduced the period of expression of osteogenic genes by the HPdLFs and, hence, accelerated model construction.

Stretch forces are common forces exerted on PDL tissues. They occur during normal chewing activity and orthodontic treatment (*Zhong et al., 2008*). Stretching forces induce relative gene changes in HPdLFs, including alterations in the expression of osteoprotegerin, receptor activator of NF-kappa B ligand, *OCN*, *COL-3*, and *Runx-2* (*Oortgiesen et al., 2012*; *Tsuji et al., 2004*; *Yu et al., 2013*). *COL-3* is a fundamental component of PDL tissue; its expression is a key marker for the formation of PDL (*Liao et al., 2013*). In this study, a stable increase occurred in the *COL-3* gene in both PN and PF groups during the 7-day stretching period. The PF group had significantly higher *COL-3* expression on day 5, and on day 7 after application of stretching forces, compared with the PN group. PDL tissue has a complex fibrillary architecture that is necessary to withstand the strong forces at the tooth-bone interface (*De Jong et al., 2017*). A complete *in vitro* model of PDL behavior must include a cellular arrangement with polarity (*Jiang et al., 2016*). The cellular polarity of HPdLFs under continuous cyclic stretch indicated the maturity of the PDL tissue (*Matsugaki, Fujiwara & Nakano, 2013*; *Oortgiesen et al., 2012*; *Papadopoulou et al., 2017*; *Wu et al., 2017*). We found that the group PF HPdLFs gained polarity and became almost parallel on day 3; this response was faster than the group PN response (day 5). These results suggested that an acceleration in tissue maturation occurred in the group PF cells. After gaining polarity, both groups maintained it until day 7, which suggested that this model can maintain maturity until at least day 7 (consistent with the PCR result). *FGF-2* is an important growth factor that enhances PDL regeneration (*Momose et al., 2016*). Both PN and PF groups showed similar increase and decrease patterns for *FGF-2* and *OCN*, but without statistically significant differences under most circumstances. This result indicated that these two types of models have a similar gene expression pattern that mimics PDL. The result of another study was similar; the expression of *OCN*, *ALP*, and *Runx-2* of tension-treated HPdLFs increased faster compared with the non-tension-treated group (*Shen et al., 2014*). Another study also reported cellular polarity and gene expression change after cyclic stretching (*Wu et al., 2017*). In their 0.5 to 24 h experiment duration, the time-course gene expression change was not significant. In our research, day 0 and day 3 gene expression showed significant difference between PF and PN groups (Fig. 5), indicating an accelerated model construction. In other reports of PDL stretching model, researchers find the rapid expression of MAPK passway; however, they used fibronectin only to coat their culturing chamber (*Papadopoulou et al., 2017*). According to our findings, using plasma and fibronectin both may accelerate the establishment of their model in a even faster speed.

## Other physical and chemical features

Although there were differences in the surface C proportions, O proportions, and wettability among the three groups other than group PN (NN, FN, and PF), none were statistically significant. Therefore, after fibronectin coating, chamber surface wettability in the plasma-treated and plasma-untreated groups became similar to that of group NN. This result

indicated that fibronectin coating did not increase PDMS chamber wettability. The PF chambers developed a fibronectin coating on the plasma-treated PDMS, but had much lower Si-($O_2$) peak values than the PN chambers (Fig. 4). The fibronectin coating had likely successfully changed the chambers' surface components to the original hydrophobic type by "pulling" low molecular weight methyl groups from the internal surface back to the outer surface of the PDMS chamber, to reduce free energy (*Larson et al., 2013*). As this phenomenon occurs, some of the surface area may remain hydrophobic and some may revert to hydrophilic, consequently forming a mosaic of high- and low-contact-angle functionality on the PDMS surface. If the sections of hydrophobicity and hydrophilicity were small enough, the cloud-like surface observed in PF chambers in the SEM images would be created. The SEM images revealed that the FN chambers had the greatest numbers of grooves and peaks of all types, and the control, untreated group had the second highest. The similarity in XPS results between these two groups suggested that the fibronectin coating could change the PDMS surface component to a very limited degree (*Kobayashi et al., 2018*). The PDMS surface was oxidized with plasma and the fibronectin coating contributed to a type of oxidation–reduction reaction for the PDMS chamber.

## Other approaches

Other polymers have been used as *in vitro* PDL-like tissue scaffold materials, including poly(lactic acid), poly(lactic-co-glycolic acid), and poly ($\varepsilon$-caprolactone) (*Ko et al., 2015*; *Liao et al., 2016*; *Liao et al., 2013*). Most polymers require some degree of surface modification to increase their biocompatibility (*Li et al., 2016*). Many kinds of surface modification methods have been reported, with protein coating or plasma discharging treatments most commonly used (*Mussig et al., 2010*; *Oortgiesen et al., 2012*; *Tsuruga et al., 2009*; *Yu et al., 2013*); however, few researchers have examined the combination of protein coating and plasma discharging. *Hattori, Sugiura & Kanamori (2010)* combined chemical and physical treatments, but their chemical treatment was based on UV-light illumination. We combined fibronectin and vacuum plasma to modify the PDMS surface. The cellular adhesive ability of the plasma vacuum-treated group (PN) was similar to that of the combination group (PF); however, the accelerated bone metabolism-related gene expression and cellular polarity appearance in group PN were indicative of accelerated model construction ability in stretch force-loaded and -unloaded conditions.

Our method can reduce the time required for human PDL tissue model construction *in vitro* and thus increase research efficiency. It has potential for use in both basic research and drug testing. If needed, various kinds of growth factors can be added to improve its versatility for use in periodontal regeneration.

## CONCLUSIONS

Construction of an *in vitro* model of human PDL tissue can be accelerated by combining vacuum plasma and fibronectin coating to a PDMS matrix. This acceleration, especially in osteogenetic gene expression, occurs both with and without the application of cyclic stretch force.

## ACKNOWLEDGEMENTS

We thank Dr. Naotaka Kishimoto from the Department of Anesthesiology, Osaka Dental University, for his valuable discussion. We also thank Mr. Hideaki Hori from the Institute of Dental Research, Osaka Dental University, for his kind help with experimental technique.

### Funding

This research was supported by the National Natural Science Foundation of China (No. 31600752 and No. 81700941) and the Sichuan University-Luzhou City cooperation project (No. 2018CDLZ-14). The funders had no role in study design, data collection and analysis, decision to publish, or preparation of the manuscript.

### Grant Disclosures

The following grant information was disclosed by the authors:
National Natural Science Foundation of China: 31600752, 81700941.
Sichuan University-Luzhou City cooperation project: 2018CDLZ-14.

### Competing Interests

The authors declare there are no competing interests.

### Author Contributions

- Wen Liao conceived and designed the experiments, performed the experiments, analyzed the data, prepared figures and/or tables, authored or reviewed drafts of the paper, approved the final draft.
- Yoshiya Hashimoto and Zhihe Zhao conceived and designed the experiments, approved the final draft.
- Yoshitomo Honda conceived and designed the experiments, performed the experiments, contributed reagents/materials/analysis tools, approved the final draft.
- Peiqi Li conceived and designed the experiments, performed the experiments, approved the final draft.
- Yang Yao conceived and designed the experiments, authored or reviewed drafts of the paper, approved the final draft.
- Naoyuki Matsumoto conceived and designed the experiments, contributed reagents/materials/analysis tools, approved the final draft.

### Data Availability

The raw measurements are available in the Supplemental Files.

### Supplemental Information

Supplemental information for this article can be found online at http://dx.doi.org/10.7717/peerj.7036#supplemental-information.

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
