# Peer review of "Accelerated construction of an in vitro model of human periodontal ligament tissue: vacuum plasma combined with fibronectin coating and a polydimethylsiloxane matrix"

_PeerJ, doi:10.7717/peerj.7036_

## Round 0.1 · original submission · Minor Revisions

Please address critiques of all reviewers and revise manuscript accordingly.



Reviewer 1 ·

Basic reporting

The manuscript shows clear and professional organization and sufficient reference and background. All the data were analyzed and interpreted appropriately. The figures and tables are shown with good quality and support the hypotheses.

Experimental design

Methods described in the manuscript is sufficient to follow. Research question well defined, relevant and meaningful.

Validity of the findings

After systematic evaluation, the authors successfully developed novel construction of an in vitro model of human periodontal ligament tissue by using vacuum plasma combined with fibronectin coating and a polydimethylsiloxane matrix. This new method is absolutely suitable for the publication. Data was interpreted completely and supported the conclusion.

Additional comments

Overall, the manuscript contains sufficient data to support its conclusion. The authors created a new in vitro model of human periodontal ligament tissue successfully. The organization of the full text is scientific and professional for publication. The authors may need to consider polishing the language for a better understanding of international researchers.

Reviewer 2 ·

Basic reporting

The authors have examined the role of the combination of fibronectin coating and
vacuum plasma treatment with polymethylsiloxane cell culture in achieving
accelerated osteogenic gene expression and construction of an in vitro human
periodontal ligament fibroblast stretching model.

General comments:

1. The manuscript is well written with a clear introduction and extensive review of the published literature.
2. The authors have brought out the research question into purview in a lucid manner.
3. The correct provenance of the cell line has been described. The methods and the technical details have been well described.

Experimental design

No comment

Validity of the findings

My major concern is with the data analysis and its representation


1. The statistics section in this MS seems to be highly inadequate where there is no clarity on the statistical tests used (paired or unpaired tests), the distribution of data and whether a post hoc analysis was done or not.
2. There is no mention of the statistical platform (Stata, R, SPSS etc.) used for the statistical analysis. The statistical paltform should be mentioned in the method section.
3. The data has been inadequately represented in the MS with the use of only p-values for statistical significance ignoring the importance of confidence intervals. Author should include confidence intervals
4. Tables for the results need to be added and the quality of the graphs should be improved.

Reviewer 3 ·

Basic reporting

A. The English language should be improved for a clearer understanding of the international audience. Manuscript revision by native English-speaking colleagues could be helpful in this regard. Some examples where languages could be improved: line396-398, 400-401, 409-410, etc. Other minor corrections are below,
- Line 62-65: Two subsequent sentences were started with “During orthodontic treatment,…”. Please consider rewriting these sentences.
- Line 80: “Chemical and physical treatment techniques…” - - > “Several chemical and physical treatment techniques…”
- Line 82: “It enhances binding of the …” - - > “It enhances the binding of the …”
- Line 84: “physical treatment technique.” - - > “physical treatment technique to enhance hydrophilicity.”
- Line 85-86: “followed by acrylic acid…” - - > “followed by treatment with acrylic acid…”
- Line 254-255: “Cell nuclei stained blue and the cytoplasm (cytoskeleton) stained green.” - - > “Cell nuclei were stained blue and the cytoplasm (cytoskeleton) were stained green.”
- Line 303-304: “PDMS is a hydrophilic material, and therefore treatment devised to improve its surface wettability is usually considered conducive to cell and extracellular matrix growth.” - - > “PDMS is a hydrophobic material, and therefore it’s chemical and physical treatments were devised to improve its surface wettability which is usually considered conducive to cell and extracellular matrix growth.”
- Line 390: “…of surface modification methods,” - - > “…of surface modification methods reported,”
- Line 397: “…cellular polarity appearance indicated accelerated model construction ability,” - - > “…cellular polarity appearance indicated the accelerated model construction ability,”

B. The introduction could be improved as suggested below,
- The main theme of the paper is the accelerated construction of an in vitro model of human periodontal ligament tissue. Please discuss the significance of developing in vitro model of human periodontal ligament tissue at the end of first paragraph (line 73) or the beginning of second (line 75) paragraph. As these types of models were reported previously (Liao et al 2013, Liao et al 2013,), please explain the context that the current model is addressing. A brief discussion about the previous models will be appropriate.
- Line 34: Sentence “Orthodontic treatment makes one beautiful.” Orthodontic treatment is used for prevention of teeth disease and beautification. As this is the introductory sentence of the abstract, a comprehensive introduction of orthodontics will be more suitable here.
- Line 62: “….tissues of tooth and bone.” Addition of a brief introduction (1-2 line) of orthodontic treatment will be helpful to the reader.
- Line 61: “….tooth roots to the alveolar bone.” Please cite the reference.
- Line 63: “….stretching and compression forces.” Please cite the reference.
- Line 72-73: “The difference between clinical observation and biochemical responses inspired us to further examine the bone metabolism mechanism in PDL tissues.” It will be clearer to the reader if rewritten as “Similar biochemical responses for directional and compression forces inspired us ….” Please elaborate on how is that observation relevant to the study presented here.
- Line 78: “…a low Young's modulus value.” Please cite the reference.
- Line 87-90: “A combination of chemical….of ECM proteins.” For clarification to the reader, please indicate that combined use of chemical and physical techniques to modify surface properties was successful before as reported in [ref].

C. Please structure the abstract as background/methods/results or any format of your choice for clarity.

D. There are few minor corrections for figures.
- Figure 4: Please provide sufficient space between E & G and F & H, respectively for more clarity. The ‘NN’ of figure E and ‘FN’ of figure F appear to be in figure G and H, respectively.
- Figure 6: “Note that cells from group PF had significantly higher Runx-2 gene expression than group PN on day 1 and day 3, and OCN gene expression on day 1.” - - > “Note that cells from group PF had significantly higher Runx-2 and OCN gene expression than group PN on day 3.” Please mention what the ‘*’ are indicating and remove the ‘*’ from day 1 (Runx-2).

E. There are few minor corrections for raw data.
- Some of the values in the raw data for Fig. 2 (AFM) are written in Chinese or Japanese. Translating these values into English will help the international audience.
- Consistent naming of the images or documents in the raw data as followed in the manuscript will improve clarity.

Experimental design

A. The current work fits well with the scope of the journal.
B. The addressed research question was well defined. As mentioned previously, the introduction needs to elaborate on how the present work will address the identified knowledge gap.
C. Experiments were performed with a high technical standard.
D. The methods are described in sufficient details to replicate the results.

Validity of the findings

A. Data is robust.
B. The conclusions are clear and adequately address the original research questions.
C. Could you please elaborate in the discussion section how the acceleration of the gene expression and cellular polarity in the current model compares with the previous models?

Additional comments

No comment.

---

## Round 0.2 · accepted · Accept

All the critiques were adequately addressed and the manuscript was revised accordingly.